# One-Pot Synthesis of Cellulose-Based Carbon Aerogel Loaded with TiO_2_ and g-C_3_N_4_ and Its Photocatalytic Degradation of Rhodamine B

**DOI:** 10.3390/nano14131141

**Published:** 2024-07-02

**Authors:** Fangqin Liu, Mingjie Fan, Xia Liu, Jinyang Chen

**Affiliations:** School of Environmental and Chemical Engineering, Shanghai University, 99 Shangda Road, Shanghai 200444, China; fqliu19@shu.edu.cn (F.L.);

**Keywords:** cellulose-based carbon aerogel, graphitic carbon nitride, photocatalysis, rhodamine B, degradation

## Abstract

A cellulose-based carbon aerogel (CTN) loaded with titanium dioxide (TiO_2_) and graphitic carbon nitride (g-C_3_N_4_) was prepared using sol–gel, freeze-drying, and high-temperature carbonization methods. The formation of the sol–gel was carried out through a one-pot method using refining papermaking pulp, tetrabutyl titanate, and urea as raw materials and hectorite as a cross-linking and reinforcing agent. Due to the cross-linking ability of hectorite, the carbonized aerogel maintained a porous structure and had a large specific surface area with low density (0.0209 g/cm^3^). The analysis of XRD, XPS, and Raman spectra revealed that the titanium dioxide (TiO_2_) and graphitic carbon nitride (g-C_3_N_4_) were uniformly distributed in the CTN, while TEM and SEM observations demonstrated the uniformly distributed three-dimensional porous structure of CTN. The photocatalytic activity of the CTN was determined according to its ability to degrade rhodamine B. The removal rate reached 89% under visible light after 120 min. In addition, the CTN was still stable after five reuse cycles. The proposed catalyst exhibits excellent photocatalytic performance under visible light conditions.

## 1. Introduction

Organic dyes are highly toxic and chemically stable, potentially teratogenic, and carcinogenic to humans. As an important dye, rhodamine B (Rh. B) is a synthetic basic cationic dye [1] widely used in industrial fields such as textiles, leather, and papermaking [2,3]. Rh. B wastewater leads to serious environmental pollution [4,5,6], may cause liver damage in humans, and is neurotoxic and carcinogenic [7,8]. Therefore, its degradation is very important.

As a “green” advanced oxidation process (AOP) technology, photocatalysis can effectively degrade organic pollutants into harmless carbon dioxide, water, mineral salts, and small-molecule organic compounds, among others, through the use of highly active substances such as hydroxyl radicals and superoxide anion radicals [9,10,11]. The process is simple and easy to implement at low cost, especially when using visible light.

As for photocatalysts, TiO_2_ is one of the most widely used [12,13,14,15,16], with good chemical and thermal stability. However, it limits the effective utilization of photogenerated electrons and holes due to disadvantages such as its photo response range in the ultraviolet region [17], wide band gap (3.0–3.2 eV), and high electron–hole recombination rate [18,19,20]. Furthermore, titanium dioxide photocatalysts easily agglomerate, making them difficult to recover after use and resulting in secondary pollution [21]. Thus, metal/non-metal doping [22,23,24,25,26,27], the development of visible light response, and surface modification have been proposed in order to improve the dispersion of TiO_2_ [28,29,30,31]. Doping metal elements can shift the bandgap and enhance visible light response [32], but some metals are toxic and prone to photocorrosion. Despite numerous TiO_2_ surface modification techniques, the efficiency of modification is limited by certain conditions [30]. Therefore, the use of nonmetallic doping or recombination such as N, S, or C is more feasible. Nitrogen-based catalysts are considered promising, which can act as electrons capture center, reducing the electron–hole recombination rate, and they have the appropriate atomic size, low ionization energy, and stability. Graphitic carbon nitride (g-C_3_N_4_) is a graphene-like layered porous organic semiconductor with a bandgap of 2.7 eV, with good visible light absorption and stability [18,33,34,35,36]. Therefore, the use of a composite of TiO_2_ and g-C_3_N_4_ can effectively improve photocatalytic efficiency.

In order to enable the recycling of suspended nanoparticle catalytic systems, immobilizing the catalyst on a porous monolithic material (e.g., graphene aerogel [37], biomass carbon aerogel [38,39,40,41], or composite film [42,43]) is a commonly used method. Cellulose-based carbon aerogels have the advantages of electron-rich structure, good stability, and abundant raw materials [44,45]. In addition, as a stable carbon-rich precursor, cellulose can provide a good three-dimensional framework [46,47,48], which can improve the efficiency of catalysts through the synergistic effects of adsorption and photocatalysis. Composite catalysts with high dispersion can be prepared by dispersing different metal salts or metal alkoxides through the carbon aerogel [49]. 

In order to improve the dispersion and recycling performance, we prepared a cellulose-based carbon aerogel loaded with a composite catalyst using a one-pot sol–gel method. In the process, the precursors of the catalytic composite and cross-linker were first dispersed into nanocellulose solution, followed by the conversion of the uniform solution to gel at a high temperature. Then, the gel was freeze-dried and carbonized to obtain the ultra-light porous composite carbon aerogel. The preparation method is simple and inexpensive. The photocatalytic efficiency, recoverability, and degradation mechanism of Rh. B with the composite catalyst were studied.

## 2. Materials and Methods

### 2.1. Materials

Refining papermaking pulp (14 wt%) was purchased from Zhejiang Transfar Group (Hangzhou, China). Urea (≥99.0%), nitric acid (HNO_3_, ≥65%), ammonium persulfate (Ap, ≥98.0%), N, N-methylene bis-acrylamide (MBA, ≥98.0%), and rhodamine B (Rh. B) were purchased from Sinopharm Chemical Reagent Co., Ltd. (Shanghai, China). Anhydrous ethanol (≥99.7%) and tert-butanol (TBA) were purchased from Aladdin (Shanghai, China). Tetrabutyl titanate (TBT, ≥99%, Adamas) and hectorite (LAPONITE RDS of BYK) were used without treatment. Edetate disodium (EDTA-2Na), 1,4-benzoquinone (BQ), Isopropanol (IPA), ethanol (C_2_H_5_OH), and sodium nitrite (NaNO_2_) were purchased from Alighting Reagent Co. (Shanghai, China). Deionized water (DI) was made at the laboratory.

### 2.2. Preparation of Composite Carbon Aerogels

Figure 1 summarizes the carbon aerogel preparation process. First, a mixture of 5 mL of tetrabutyl titanate, 10 mL of ethanol (50 wt%), and 1 mL of nitric acid was added dropwise to 20 mL of ethanol, and the titanium precursor was obtained through magnetic stirring for 3 h (800 r/min). Fine refining papermaking pulp (2 g) was dissolved in deionized water to obtain the cellulose solution (1 wt%). Ap as initiator (0.1 g) and urea were added, and the mixed solution was subjected to ultrasonic dispersion for 10 min (SONICS-1800 W, Taixing Xingjian Chemical Machinery Co., Ltd., Taixing, China). Subsequently, the hydrolyzed TBT solution (4 mL) was added and stirred for 1 h. Finally, hectorite aqueous solution (5 mL, 2 wt%) and MBA alcohol (5 mL, 2 wt%) solution were added, and the mixture was stirred for another 1 h. The uniformly mixed dispersion was poured into multiple cylindrical molds (25 mm in diameter and 15 mm in height), and hydrogels were obtained through heating at 80 °C for 2 h. Unreacted ions on the surface of the hydrogel were washed off with deionized water, and the gels were aged for 12 h.

The obtained hydrogels were immersed in ethanol (50%, 75%) and TBA (50%) for solvent replacement, and then freeze-dried in a freeze-dryer (SCIENTZ-18ND, Ningbo Xinzhi Biotechnology Co., Ltd., Ningbo, China) for 48 h to obtain the aerogels. After drying, the carbon aerogels were obtained via carbonization (3 °C/min to 600 °C and maintained for 2 h) in a tube furnace (SK-G04123K, Tianjin Zhonghuan Electric Furnace Co., Ltd., Tianjin, China) under a nitrogen atmosphere. Cellulose-based carbon aerogels, TiO_2_/cellulose carbon aerogels, and g-C_3_N_4_/TiO_2_/cellulose carbon aerogels are denoted as CCA, CTA, and CTN, respectively. The amount of added urea was 1, 2, or 3 g, and the respective carbon aerogels were labeled as CTN_1_, CTN_2_, and CTN_3_.

### 2.3. Characterization

XRD patterns were obtained using a X-ray diffractometer (Rigaku MiniFlex600, Japan) with Cu Kα radiation (45 kV, 50 mA) in the range of 2θ = 5–75°. The step length was 0.02° and the scanning speed was 2°/min. The SEM and EDS measurements were conducted using a German ZEISS Sigma 300 scanning electron microscope (15.00 kV). The transmission electron microscope images were obtained using an FEI Talos F200X G2 instrument (200 kV) from the United States(FEI Company, Hillsboro, OR, USA). XPS were measured on the American Thermofisher Nexsa (Waltham, MA, USA) by using 12 kV Al Kα X-ray radiation at 150 W. The binding energies were calibrated according to the carbonaceous C1s at 284.6 eV. XPS were fitted by Gaussian–Lorentzian convolution functions with simultaneous optimization of the background parameters. Raman spectra were determined using a Horiba LabRAM HR Evolution from Kyoto, Japan with 532 nm laser excitation at 2 mW. The N_2_ adsorption–desorption isotherms were determined using a Micromeritics ASAP 2460 instrument from the United States at liquid nitrogen temperature (77 K). Electrochemical impedance spectroscopy (EIS) was conducted using an electrochemical analyzer (CHI604E, Shanghai Chenhua Instrument Co., Ltd., Shanghai, China) with a three-electrode device (carbon aerogel loaded on a glassy carbon electrode as a working electrode, platinum sheet as a counter electrode, and saturated calomel electrode as a reference electrode).

### 2.4. Adsorption of Carbon Aerogel

A total of 4 mg of carbon aerogel was added to 5, 10, 15, 20, or 25 mg/L of 30 mL Rh. B solution, and then stirred for 30 min under dark conditions. Next, 2 mL of solution was filtered with a needle filter and the absorbance was measured at 554 nm using an ultraviolet spectrophotometer (WF Z UV-2800H, Unico, Suite E, Dayton, NJ, USA), and the concentration of Rh. B was calculated using the Rh. B standard curve drawn before the experiment (Appendix A). The adsorption capacity (*q_e_*, mg/g) was calculated using Equation (1):(1)$$q_{e}=\frac{\left(C_{0}-C_{e}\right)V}{m},$$
where *C*_0_ (mg/L) and *C_e_
*(mg/L) represent the initial concentration of the Rh. B solution and the concentration after 30 min, respectively; *V* is the volume of the Rh. B solution (L); and m is the mass of the carbon aerogel (g).

### 2.5. Photocatalytic Degradation

The photocatalytic activity of the composite catalyst was evaluated through measuring the removal efficiency of Rh. B under visible light radiation (Xenon lamp, GXZ500, Shanghai Jiguang Special Lighting Appliance Factory, Shanghai, China) with 1 M NaNO_2_ as a 400 nm cut-off filter solution [50]. The 10 mg composite carbon aerogel catalyst was added to a beaker with 40 mL, 10 mg/L of aqueous Rh. B solution. After magnetic stirring for 30 min in the dark, it was irradiated 20 cm away from the light source. The absorbance of about 2 mL of suspension filtered using a 0.45 μm filter membrane was measured at 554 nm using a UV–vis spectrophotometer (WF Z UV-2800H, Unico, Suite E, Dayton, NJ, USA). In addition, the absorption spectra of Rh. B were measured every 30 min under CTN_2_ degradation using a TU-1901 UV-vis spectrophotometer in the wavelength range of 450–650 nm (Beijing PUXI General Instrument Co., Ltd., Beijing, China). As for cyclic degradation, the Rh. B adsorbed on the carbon aerogel was completely degraded under light radiation, and the carbon aerogel was centrifuged and dried (120 °C, 2 h) for further use.

### 2.6. Radical Inhibitors Experiment

In the process of a radical inhibition experiment, EDTA-2Na (5 mM), BQ (5 mM), and IPA (40 mM) were employed as scavengers of holes (h^+^), superoxide free radicals (·O_2_^−^), and hydroxyl free radicals (·OH), respectively, and other experimental conditions were similar to those of the photocatalytic degradation (40 mL, 10 mg/L Rh. B solution and 10 mg CTN_2_ samples).

## 3. Results and Discussion

### 3.1. Morphology

The electronic photographs of the aerogel and carbon aerogel clearly demonstrate that the carbonized composite maintained a three-dimensional structure (Appendix A), and the average density of the carbon aerogel was only 0.0209 g/cm^3^, indicating its ultra-light and low-density characteristics.

Figure 2a–d show the surface morphology of the CTN sample, indicating that the carbon aerogel has a rough surface and porous structure. From Figure 2f, it can be concluded that the elements C, N, O, and Ti are uniformly distributed in the material and have low aggregation, while Appendix A shows the well-dispersed nanoparticles. In addition, transmission electron microscopy (TEM) images of CTN reveal lattices with a spacing of 0.340, 0.210, and 0.207 nm [51], which may correspond to the (204) and (004) planes of TiO_2_, and the (001) plane of g-C_3_N_4_ (Appendix A), respectively. The results indicate that cellulose can be used as an excellent support catalyst through increasing the reaction area, improving the adsorption and degradation capacity, and facilitating recycling.

### 3.2. Structure and Composition State

The XRD pattern is shown in Figure 3, and the wide peak at 20–25° in the XRD pattern of CCA indicates that the material has a disordered structure due to the strong interaction of materials during the synthesis process [52]. This amorphous structure makes it a suitable material for fixing TiO_2_ and enhancing the catalytic properties of the system. In the cellulose-based carbon aerogel loaded with TiO_2_ (CTA), the characteristic diffraction peaks of TiO_2_ anatase are 25.5, 37.6, 48.3, 54.1, and 63.2°, corresponding to the crystal planes (101), (004), (200), (105), and (204), respectively [53]. This indicates that the TiO_2_ was fixed to the carbon aerogel, while the relatively narrow and sharp peaks indicated that the TiO_2_ crystallized well. The characteristic diffraction peaks of graphitic carbon nitride (g-C_3_N_4_) were observed to be 32.2° and 37.3°, corresponding to the crystal planes (200) and (001), which is in agreement with the card JCPDS no. 78-1691 and thus indicates no change the crystalline properties of TiO_2_.

XPS can be used to analyze the chemical valence state of elements. The XPS spectrum of g-C_3_N_4_/TiO_2_/CCA is shown in Figure 4a. The peaks at 284.68, 400.75, 459.27, and 532.81 eV correspond to C 1s, N 1s, Ti 2p, and O 1 s, respectively. The narrow scan spectrum of Ti 2p is shown in Figure 4b, where Ti^4+^2p_3/2_, Ti^3+^2p_3/2_, Ti^3+^2p_1/2_, and Ti^4+^2p_1/2_ correspond to 459.3, 459.9, 464.9, and 465.7 eV [54], respectively, indicating that Ti^4+^ is the main valence state. The narrow scanning spectra of N1s in Figure 4c show the binding energy peaks of C=N-C, N-(C_3_), and C-NH_x_ at 398.6, 400.6, and 402.4 eV [55], respectively, indicating the incorporation of g-C3N4 into the aerogel. 

The Raman spectroscopy results are shown in Figure 4d, where the D peak at 1350 cm^−1^ and the G peak at 1580 cm^−1^ can be used to understand the defects and crystallinity of carbon materials [56,57]. With the load with the catalyst, the increase in I_D_/I_G_ of the carbon aerogel indicates an increase in carbon disorder defects, which demonstrates that the catalyst can strip the carbon structure of the aerogel during the carbonization process, resulting in carbon defects or layer barriers. The increase in carbon defects may be due to the doping of nitrogen and titanium atoms, resulting in the absence of carbon atoms, as well as the formation of new chemical bonds between nitrogen, titanium, and carbon atoms, leading to carbon–carbon bond breakage and the reorganization of sites [58], which suggests that the catalysts are not simply physically mixed in the cellulose backbone. This increase in defects exposes more active sites and contributes to the improvement in photocatalytic activity and stability.

The N_2_ adsorption–desorption isotherms of cellulose aerogels and carbon aerogels loaded with catalysts were found to be type IV isotherms, which indicates typical mesoporous materials and corresponds to porous structural characteristics (Figure 5). The pore size distribution curves of the proposed aerogels show that the pore size was mainly distributed in the range from 10 to 20 nm, with an average pore size of 12 nm, while the pore size of the carbon aerogel was mainly distributed between 2 and 10 nm, with an average pore size of 7 nm. The specific surface areas calculated using the BET method were 14 m^2^/g and 23 m^2^/g, respectively. The abundant mesopores and relatively large specific surface area of carbon aerogels can provide sufficient adsorption and photocatalytic sites for photocatalytic degradation.

The decomposition and transport characteristics of photogenerated charge carriers were further analyzed through electrochemical impedance spectroscopy (EIS). It is well-known that the smaller the arc radius in the Nyquist diagram, the lower the resistance [59]. The Nyquist plots of EIS spectra for all four CTN samples showed that CTN_2_ had a smaller arc radius than CTN_0_. Figure 6 indicates that the incorporation of carbon nitride improved the charge transfer efficiency, which can be attributed to the internal electric field generated at the carbon aerogel interface, thus accelerating electron transfer in the bilayer. The formation of internal electric fields can be attributed to the catalysts forming a heterogeneous structure in the carbon aerogel, thus altering the internal charge distribution in the carbon aerogel. Furthermore, titanium dioxide, carbon nitride, and carbon aerogel have different electron affinities, which creates a charge shift or polarization effect within the material [60]. These internal electric fields reduce the resistance of the composite carbon aerogel, thus improving its electrical properties. In addition, the amount of carbon nitride also affects the efficiency of charge transfer separation. The increase in resistance of CTN_3_ compared with CTN_2_ may be due to the uneven dispersion caused by excessive urea addition during the synthesis process. The minimum resistance of CTN_2_ was preliminarily verified to indicate the best photocatalytic activity in the samples.

### 3.3. Photodegradation of Rh. B

As shown in Figure 7a, the saturated adsorption capacity of the carbon aerogels was 83.16 mg/g, 99.36 mg/g, and 126.28 mg/g, respectively, considering the Langmuir isothermal model to elucidate the adsorption process of rhodamine by the carbon aerogels (Appendix A). 

Figure 7b depicts the photodegradation performance of different carbon aerogels for 10 mg/L Rh. B. It can be seen that, with no catalyst, the removal rate of the blank group was 8.4% after 120 min, which may be due to the photosensitization of Rh. B. The group containing the catalyst began to illuminate after 30 min of dark reaction. After 120 min, the removal rate of Rh. B was only 52.5% in the carbon aerogel only loaded with TiO_2_, while the removal rate for the carbon aerogels loaded with g-C_3_N_4_ and TiO_2_ was 60.4%, 64.1%, and 88.1%, respectively. The results show that g-C_3_N_4_ and TiO_2_ can synergistically degrade Rh. B and improve the photocatalytic activity of the aerogel. 

The degradation kinetics of Rh. B were fitted using Equation (2) as the pseudo-first-order model—where *C*_0_ is the initial concentration of the solution, *C* is the concentration at a different time, *k* is the reaction rate constant (min^−1^), and *t* is the time (min)—as shown in Figure 7c. All data fit the pseudo-first-order kinetic model well, with CTN_2_ exhibiting the highest first-order rate constant of 0.0153 min^−1^, being 2.83 times higher than that of CTN_0_ (*k* = 0.0054 min^−1^).
(2)$$ln\left(\frac{C}{C_{0}}\;\right)=-kt$$

In order to test the reusability and stability of carbon aerogels, Figure 7d shows that the removal rate of Rh. B by CTN_2_ was 84% after five cycles of degradation, indicating that the proposed carbon aerogels have excellent recycling performance.

### 3.4. Mechanism of Photocatalytic Degradation

To investigate whether Rh. B degradation is photocatalytic or self-photosensitized, the absorption spectra of Rh. B were examined during the degradation process (Figure 8a). With the rapid decrease in the maximum absorption peak, the maximum absorption wavelength was only slightly blue-shifted, indicating that the ring-opening reaction of benzene plays a dominant role in the degradation process, rather than N-dealkylation [61,62], suggesting that the degradation of Rh. B is mainly due to photocatalysis. 

In order to further investigate the electron migration pathway of the carbon aerogels during photocatalysis, EDTA-2Na, BQ, and IPA were employed as scavengers of holes (h^+^), superoxide free radicals (·O_2_^−^), and hydroxyl free radicals (·OH), respectively, for radical capture experiments. As can be seen from Figure 8b, the capture of h^+^, ·O_2_^−^, and ·OH resulted in decreased degradation efficiencies of 68%, 49%, and 82%, respectively, indicating that the degradation process was primarily associated with ·O_2_^-^ and h^+^, consistent with previous findings [62].

Based on the above analysis and previous literature reports [63,64], the possible mechanism for the CTN-mediated decomposition of Rh. B is proposed. As shown in Figure 9a, the degradation process of Rh. B primarily encompasses four sequential steps: de-ethylation, chroma degradation, ring opening, and mineralization. In Figure 9b, the three-dimensional porous structure of the cellulose carbon aerogel ensures that the pollutant Rh. B is in full contact with the catalyst, following which the electrons are excited from the valence band of TiO_2_ to the vacant oxygen position and Ti^3+^ under light conditions with g-C_3_N_4_ to the conduction band at the same time [65]. The photogenerated electrons tend to migrate from the vacant oxygen position and Ti^3+^ of TiO_2_ to the valence band of g-C_3_N_4_ and the electron-rich surface of g-C_3_N_4_, which causes the charge in the composite system to effectively separate and produce more active free radicals (·OH), thus improving the degradation efficiency of pollutants. The reaction equations are as follows:(3)$${TiO}_{2}+g-C_{3}N_{4}\overset{hv}{\to }e^{-}+h^{+}$$
(4)$$e^{-}+O_{2}\to ∙O_{2}^{-}$$
(5)$$h^{+}+H_{2}O\to ∙OH$$
(6)$${∙O}_{2}^{-}+h^{+}+Rh.B\to Inter+{CO}_{2}+H_{2}O$$

## 4. Conclusions

A novel carbon aerogel loaded with TiO_2_ and g-C_3_N_4_ was synthesized using a one-pot method. In the gelation process, hectorite was used as a cross-linking agent to enhance the three-dimensional network, and after carbonization, the in situ synthesized carbon aerogel maintained a porous structure and may possess more surface functional groups. Due to the synergistic effects of g-C_3_N_4_ and TiO_2_, the composite catalyst had a high utilization rate of visible light and uniform dispersion, and the removal rate of Rh. B reached 88.1% after 120 min. The degradation efficiency was not significantly reduced after five reuse cycles, indicating that the proposed carbon aerogel has good catalytic activity and reusability.

## Figures and Tables

**Figure 1 nanomaterials-14-01141-f001:**
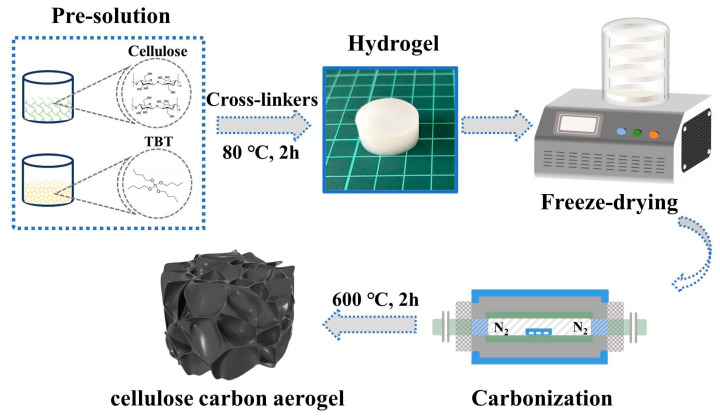
Schematic graph of carbon aerogel preparation.

**Figure 2 nanomaterials-14-01141-f002:**
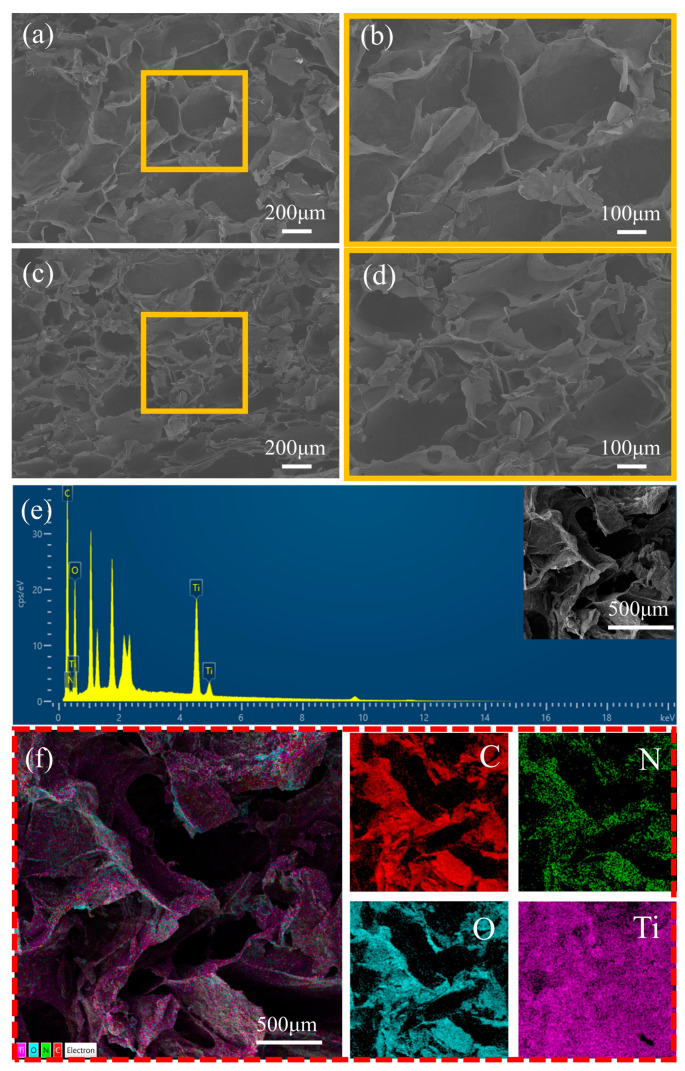
SEM images of CTN: (**a**,**c**) cross-section of the CTN sample; (**b**,**d**) corresponding enlarged view; (**e**) EDX spectrum of CTN sample; and (**f**) element mappings of C, N, O, and Ti.

**Figure 3 nanomaterials-14-01141-f003:**
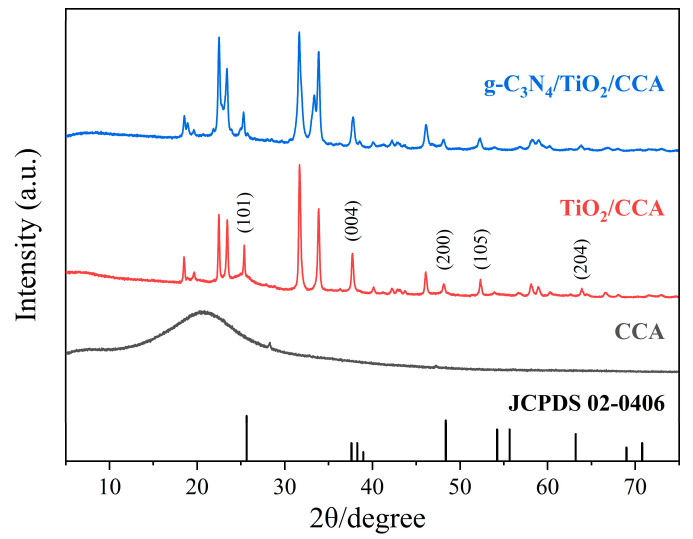
XRD patterns of CCA, TiO_2_/CCA, and g-C_3_N_4_/TiO_2_/CCA samples.

**Figure 4 nanomaterials-14-01141-f004:**
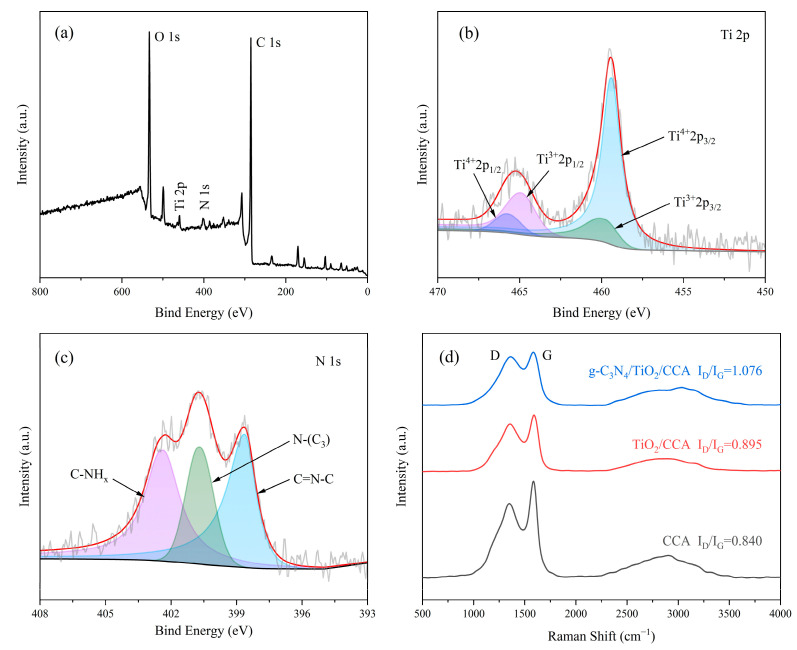
XPS spectrum of g-C_3_N_4_/TiO_2_/CCA: (**a**) full spectrum; (**b**) Ti 2p; and (**c**) N 1s. (**d**) Raman spectra of CCA, TiO_2_/CCA, and g-C_3_N_4_/TiO_2_/CCA samples.

**Figure 5 nanomaterials-14-01141-f005:**
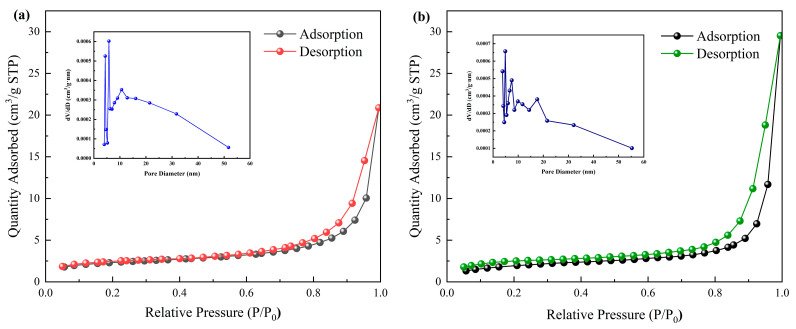
(**a**) N_2_ adsorption–desorption isotherm of g-C_3_N_4_/TiO_2_ cellulose aerogel; (**b**) N_2_ adsorption–desorption isotherm of carbon aerogel; illustration: respective pore size distribution curves.

**Figure 6 nanomaterials-14-01141-f006:**
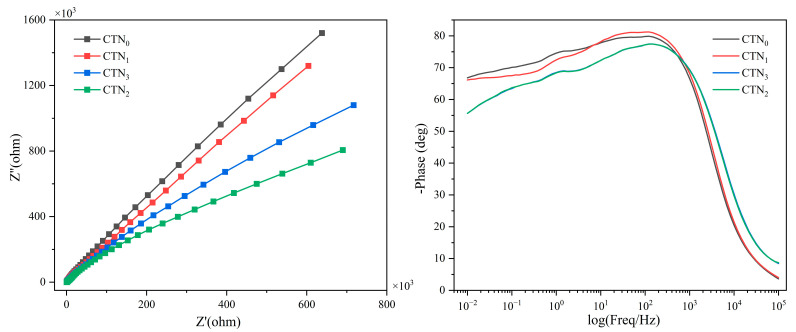
Nyquist and Bode-phase plots of CTN samples.

**Figure 7 nanomaterials-14-01141-f007:**
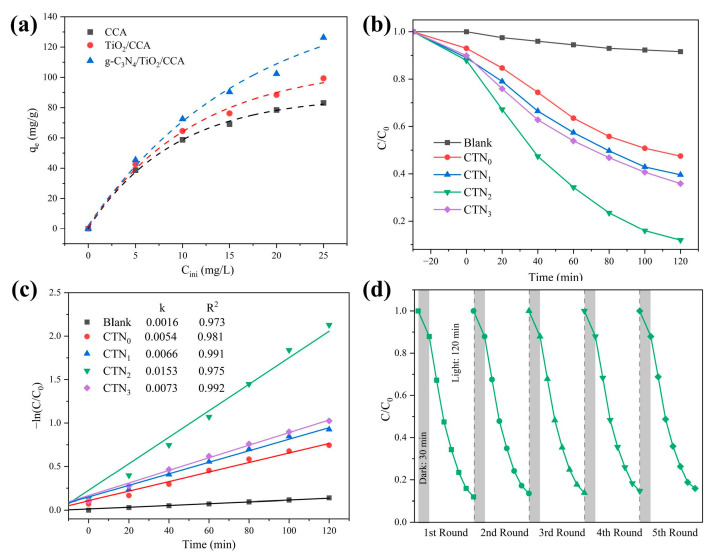
(**a**) The isothermal Rh. B adsorption curves of CCA, TiO_2_/CCA, and g-C_3_N_4_/TiO_2_/CCA samples; (**b**) the degradation curves of Rh. B on carbon aerogels; (**c**) the relevant reaction kinetics of decomposing Rh. B; and (**d**) the cyclic degradation of CTN_2_.

**Figure 8 nanomaterials-14-01141-f008:**
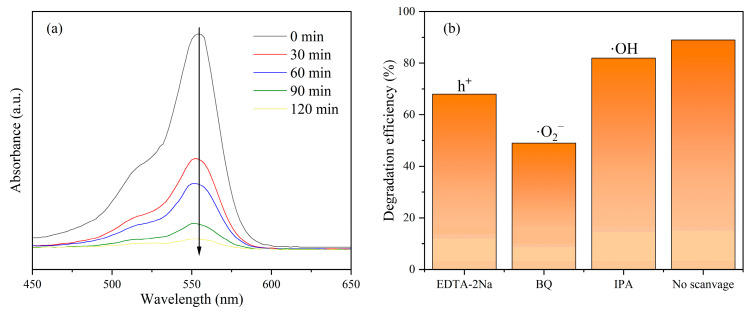
(**a**) Absorption spectra of Rh. B under g-C_3_N_4_/TiO_2_/CCA degradation; and (**b**) degradation efficiency of g-C_3_N_4_/TiO_2_/CCA under the condition of active substance capture.

**Figure 9 nanomaterials-14-01141-f009:**
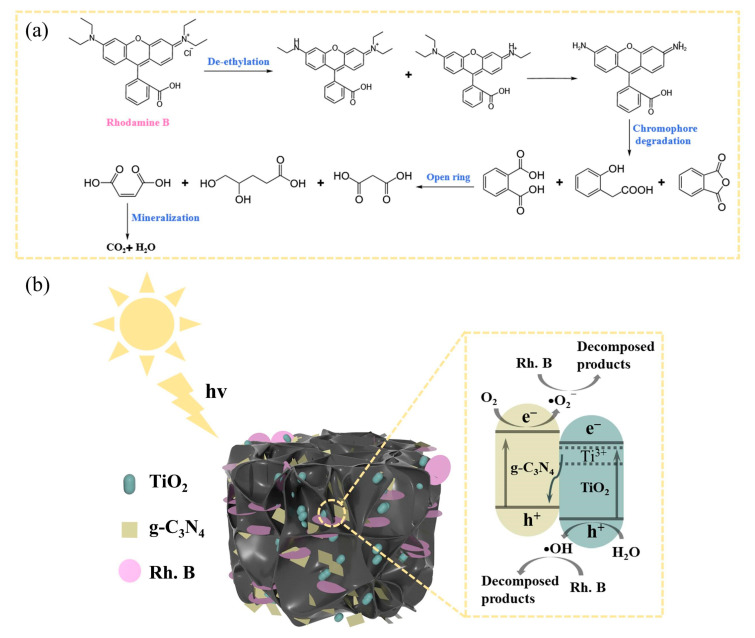
(**a**) Possible mechanisms of Rh. B degradation; and (**b**) schematic diagram of photodegradation of Rh. B by CTN.

## Data Availability

Data are unavailable due to privacy and ethical restrictions.

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
