# Peer review of "One-Pot Synthesis of Cellulose-Based Carbon Aerogel Loaded with TiO2 and g-C3N4 and Its Photocatalytic Degradation of Rhodamine B"

_nanomaterials, 2024, doi:10.3390/nano14131141_

Round 1

Reviewer 1 Report

Comments and Suggestions for Authors

The present work describes synthesis and functional properties of a cellulose aerogel loaded with titanium oxide. This type of photocatalysts is extensively studied. Unfortunately, the manuscript does not provide any context and comparison with similar works, such as [10.1007/s13204-021-02328-y],[10.1088/1361-6528/acf9ad] and many others. The work is written in poor English, which significantly complicates the understanding. Nouns are frequently used instead of verbs.

The experimental section lacks details. For every method the exact experimental conditions must be added. For XRD – step size and accumulation time; for SEM – type of detector, accelerating voltage; for BET - pressure values, etc. In the experimental section it is not described how isotherms of nitrogen sorption/desorption was obtained. IR spectroscopy might be useful for the characterisation of the obtained material as well.

The choice to place some images in the ESI is questionable. The Results and Discussion section (line 132) starts very abruptly, there is no discussion of the original aerogel,   the original porous structure is not discussed.

The text includes a lot of information which is better to place in the images, for example the hkl indexes in line 152. It is more readable in the picture and reiterating this information in the text adds no value to the discussion. Similar problem is with description of other methods as well. 

In line 189-190 pore size is given as 12.0853 nm, which is absurd. In line 192 surface area determined by BET is given with unrealisctic precision. This reveal poor understanding of the method.

The suggested mechanism of rhodamine B degradation is not based on any experimental work. Commonly chemiluminescent ROS sensors are used to eveluate the ROS in the system.

In the current state I can not recommend the work for publication. It requires some additional experiments and the text needs to be modified significantly.

Comments on the Quality of English Language

he work is written in poor English, which significantly complicates the understanding. Nouns are frequently used instead of verbs.

Author Response

Thank you very much for taking the time to review this manuscript. The necessary adjustments have been incorporated into the resubmitted version.

comments 1: The present work describes synthesis and functional properties of a cellulose aerogel loaded with titanium oxide. This type of photocatalysts is extensively studied. Unfortunately, the manuscript does not provide any context and comparison with similar works, such as [10.1007/s13204-021-02328-y],[10.1088/1361-6528/acf9ad] and many others. The work is written in poor English, which significantly complicates the understanding. Nouns are frequently used instead of verbs.

Response 1: We agree with this comment, so we have added references to the introduction to provide more background information. We are aware of the deficiencies in the language, so the manuscript has been revised and corrected based on professional advice.

comments 2: The experimental section lacks details. For every method the exact experimental conditions must be added. For XRD – step size and accumulation time; for SEM – type of detector, accelerating voltage; for BET - pressure values, etc. In the experimental section it is not described how isotherms of nitrogen sorption/desorption was obtained. IR spectroscopy might be useful for the characterisation of the obtained material as well.

Response 2:  Thank you for pointing this out. We added specific experimental conditions in the experimental part, such as adding XRD step size and accumulation time (line 105), describing the detector type in line 106, and providing the conditions for obtaining nitrogen adsorption/desorption isotherms in line 111.

comments 3:  The choice to place some images in the ESI is questionable. The Results and Discussion section (line 132) starts very abruptly, there is no discussion of the original aerogel,   the original porous structure is not discussed.

Response 3: Sample image was placed in the ESI image for neat typography. Changes have been made to line 140 of the Results and Discussion section.

comments 4: The text includes a lot of information which is better to place in the images, for example the hkl indexes in line 152. It is more readable in the picture and reiterating this information in the text adds no value to the discussion. Similar problem is with description of other methods as well. 

Response 4:  We apologize for restating specific details in the XRD and XPS results discussion. Our aim was to enhance clarity and conciseness while ensuring comprehensive coverage of information.

comments 5:  In line 189-190 pore size is given as 12.0853 nm, which is absurd. In line 192 surface area determined by BET is given with unrealisctic precision. This reveal poor understanding of the method.

Response 5:  We have modified the specific surface area and pore size accuracy determined by BET, such as 12.09 nm (line 198).

comments 6:  The suggested mechanism of rhodamine B degradation is not based on any experimental work. Commonly chemiluminescent ROS sensors are used to eveluate the ROS in the system.

Response 6:  Thanks for pointing it out. Due to limited experimental conditions and time constraints, we conducted radical trapping experiments, detected the ultraviolet absorption spectrum of Rh. B during the degradation process, and referred to the reported research work to assist in the reasoning the possible degradation mechanism of Rhodamine. The part of the mechanism has been revised (line 251).

The grammar and linguistic expression of the full text have been revised in accordance with the feedback from the journal's professionals. Thank you very much for your valuable advices.

Reviewer 2 Report

Comments and Suggestions for Authors

The authors talk about the study of photocatalytic activity of carbon aerogels loaded with TiO2 and g-C3N4, however, there are some requests or commend should be considered:

1.     In this study, Ph. B was used for the photocatalytic activity test, however, dye sensitization could occur with photocatalysts, on the other hand, color of Rhodamine B could be faded under UV/vis irradiation.  Therefore, the use of Rhodamine B as a target pollutant in this study does not quite meet the requirements of photocatalytic experiments.

2.     The references for explanation of “TiO2 is one of the most widely used”, “metal/non-metal doping”, “surface modification” are not enough. Please find, read and study more references. ----Line 37, Line 42, Line 43 and Line 64.

3.     Focus on the book of “The ACS style Guide”, in chapter 11: “Use numerals with units of time or measure and use a space between the numeral and the unit.”.  So, “20ml” should be “20 ml”, “2g” should be “2 g”. Please modify all of them in this manuscript.

4.     “Rh B” should be “Rh. B”, “Rh.” is short for “Rhodamine”. Please understand what “Rhodamine” is.

5.     Line 139: “0.340, 0.210, and 0.207” should be “0.340, 0.210 and 0.207”.

Author Response

Thank you very much for taking the time to review this manuscript. The necessary adjustments have been incorporated into the resubmitted version.

Comments 1: In this study, Ph. B was used for the photocatalytic activity test, however, dye sensitization could occur with photocatalysts, on the other hand, color of Rhodamine B could be faded under UV/vis irradiation.  Therefore, the use of Rhodamine B as a target pollutant in this study does not quite meet the requirements of photocatalytic experiments.

Response 1: We agree with this comment, referring to other reports of photocatalytic degradation of Rh. B and according to the blank experimental results of this study, the photosensitization is less than 10% (line 230), while the absorption spectra of Rh. B is slightly blue shifted (line 254), indicating that oxidation ring-opening reaction is dominant, rather than photosensitization.

Comments 2: The references for explanation of “TiO2 is one of the most widely used”, “metal/non-metal doping”, “surface modification” are not enough. Please find, read and study more references. ----Line 37, Line 42, Line 43 and Line 64.

Response 2: Thanks for pointing it out. We have added some new references in line 39 " TiO2 is one of the most widely used", line 44 "metal/non-metal doping" and line 46 "surface modification".

Comments 3: Focus on the book of “The ACS style Guide”, in chapter 11: “Use numerals with units of time or measure and use a space between the numeral and the unit.”.  So, “20ml” should be “20 ml”, “2g” should be “2 g”. Please modify all of them in this manuscript.

Response 3: Thank you so much for pointing it out. We have revised it all in the manuscript (line 81, 82).

Comments 4: “Rh B” should be “Rh. B”, “Rh.” is short for “Rhodamine”. Please understand what “Rhodamine” is.

Response 4: Thanks for pointing it out. The "Rh B" in this manuscript has been changed to "Rh. B".

Comments 5: Line 139: “0.340, 0.210, and 0.207” should be “0.340, 0.210 and 0.207”.

Response 5: Thanks for pointing it out. This (line 149) and the full text has been checked and modified.

Thank you again for your valuable suggestions

Reviewer 3 Report

Comments and Suggestions for Authors

The author produced a cellulose-based modified catalyst with the aim of being able to efficiently degrade the rhodamine B pollutant with visible light. After reading the article, my comments and questions are as follows:

I miss the spectrum of the light source. How was it implemented so that the irradiation only takes place with visible light?

The dye used as a model compound is colored. The photosensitivity of colored materials can modify the effect of the catalyst. How was the phenomenon of photosensitization eliminated?

It would be worthwhile to examine the activity of the catalyst also in the case of a colorless model compound.

The degradation of rhodamine B was monitored by measuring the light absorption values. Has the organic carbon content of the reaction mixture changed? Has mineralization occurred? This would be important to examine. How long would it take for complete mineralization?

Comments on the Quality of English Language

The language of the article is correct.

Author Response

Thank you very much for taking the time to review this manuscript. The necessary adjustments have been incorporated into the resubmitted version.

Comments 1: I miss the spectrum of the light source. How was it implemented so that the irradiation only takes place with visible light?

Response 1: Thank you for pointing this out. In the experimental section, we have restated the light source conditions (line 130). The lamp is placed in a double glass cover filled with 1M NaNO2 as a 400 nm cut-off filter solution.

Comments 2: The dye used as a model compound is colored. The photosensitivity of colored materials can modify the effect of the catalyst. How was the phenomenon of photosensitization eliminated?

Response 2: We agree with this comment, referring to other reports of photocatalytic degradation of Rh. B and according to the blank experimental results of this study, the photosensitization is less than 10% (line 230), while the absorption spectra of Rh. B is slightly blue shifted (line 254), indicating that oxidation ring-opening reaction is dominant, rather than photosensitization.

Comments 3: It would be worthwhile to examine the activity of the catalyst also in the case of a colorless model compound.

Response 3: We agree with this comment. This is indeed a deficiency of our research work.

Comments 4: The degradation of rhodamine B was monitored by measuring the light absorption values. Has the organic carbon content of the reaction mixture changed? Has mineralization occurred? This would be important to examine. How long would it take for complete mineralization?

Response 4: Thanks for pointing it out. In the degradation experiment, we take a few milliliters of the reaction mixture to check the absorbance change, and then put it back into the mixed system, so the catalyst content change is negligible. Mineralization occurred during the degradation reaction, but the timing of mineralization is difficult to determine.

Thank you again for your valuable suggestions

Round 2

Reviewer 1 Report

Comments and Suggestions for Authors

Dear authors!

I appreciate the time and effort you dedicated to the revision process. Below I provide some answers to your Responses, as well as some new comments:

Response 1: We agree with this comment, so we have added references to the introduction to provide more background information.

Comment to response 1: The introduction is improved, but adding references does not equal providing context and comparison. Please discuss provided references and discuss advantages and disadvantages of different approaches used to modify TiO2 charachteristics, etc. The aim of the introduction is to place the work into context of already existing research and motivate the choices used in your particular experiment.

Response 2:  Thank you for pointing this out. We added specific experimental conditions in the experimental part, such as adding XRD step size and accumulation time (line 105), describing the detector type in line 106, and providing the conditions for obtaining nitrogen adsorption/desorption isotherms in line 111.

Comment to response 2: Thank you for providing experimental details in this section. However, I provided XRD, SEM and BET just as examples. Every used method needs to be described with appropriate detail. For example, you mentioned the accelerating voltage for SEM, but not for TEM. For Raman analysis at least the laser wavelength and power must be added

Here is an example of proper XPS description:

The composition and chemical state of the elements were studied by X-ray photoelectron spectroscopy (XPS). The measurements were effectuated on a K-Alpha (Thermo Fisher Scientific, Waltham, MA, USA) spectrometer equipped with a monochromatic AlKα X-ray source (E = 1486.7 eV). The positions of the peaks in the binding energy scale were determined with respect to the C1s peak corresponding to the carbon contamination of the surface (285.0 eV) with an accuracy of 0.1 eV. XP-spectra were fitted by Gaussian-Lorentzian convolution functions with simultaneous optimization of the background parameters.

Similar level of detalization must be given for all the used methods.

Response 5:  We have modified the specific surface area and pore size accuracy determined by BET, such as 12.09 nm (line 198).

Comment to response 5: BET is not a very precise method, especially given that the surface area of the obtained samples is not extremely high. Usually decimal places are not given for BET data at all. Please see this publication for more detail https://onlinelibrary.wiley.com/doi/10.1002/adma.202201502

Response 6:  Thanks for pointing it out. Due to limited experimental conditions and time constraints, we conducted radical trapping experiments, detected the ultraviolet absorption spectrum of Rh. B during the degradation process, and referred to the reported research work to assist in the reasoning the possible degradation mechanism of Rhodamine. The part of the mechanism has been revised (line 251). 

Comment to response 6: I appreciate you conducting extra experiments to give more basis for the suggested mechanism. Please include description of this experiment in the Materials and Methods section. As of now, the reactants are mentioned, but the experiment is not described.

New comment 1: In lines 194-203 the absorption isotherms are discussed. The volume of absorbed nitrogen is merely 20-30 ml (for SiO2 aerogel this value can reach 2 l). This leads no high signal/noise ratio in the pore size distribution curves. It might be better to not provide the image in the text and just mention the cumulative pore volume. If the authors choose to keep this image, please make the y-axis be of the same scale (not 0-20 and 0-30 cm3/g, but both pictures 0-30 cm3/g) and make the caption in the insterted images with pore size distribution readable.

New comment  2: Line 208 states: “the smaller the arc radius in the Nyquist diagram, the lower the resistance”, but the radii of the experimentally obtained curves are not given anywhere and the reader is expected to make qualitative conclusions based on the image alone.

New comment 3: Line 293 states: “in situ synthesized carbon aerogel possessed 293 more surface functional groups” but the surface groups were not characterized in the work. Use of IR spectroscopy was suggested in my previous review report, but this method was not included. This is up to the Authors, but the conclusion about surface groups seems unbased this way.

Author Response

We sincerely appreciate your careful review of our manuscript and your valuable comments.

In the revised manuscript, we used red font and underline to mark the changes. Here are our responses to the comments and comments.

Comments 1: The introduction is improved, but adding references does not equal providing context and comparison. Please discuss provided references and discuss advantages and disadvantages of different approaches used to modify TiO2 charachteristics, etc. The aim of the introduction is to place the work into context of already existing research and motivate the choices used in your particular experiment.

Response 1: In the introduction part, we have added the advantages and disadvantages of the methods to improve TiO2, which serves as a smooth transition to explain why we chose graphitic carbon nitride to enhance visible light response (lines 46-53).

Comments 2: Thank you for providing experimental details in this section. However, I provided XRD, SEM and BET just as examples. Every used method needs to be described with appropriate detail. For example, you mentioned the accelerating voltage for SEM, but not for TEM. For Raman analysis at least the laser wavelength and power must be added.

Response 2: Thank you for pointing this out. In the characterization section, we have revised the testing details, such as the TEM acceleration voltage of 200kV, the Raman test wavelength of 532nm with a power of 2mW, and the type of exciter, acceleration voltage, and power for XPS, etc. (lines 116-121).

Comments 5: BET is not a very precise method, especially given that the surface area of the obtained samples is not extremely high. Usually decimal places are not given for BET data at all.

Response 5: Thank you for pointing it out. The data obtained from BET has been revised (line 218).

Comments 6: I appreciate you conducting extra experiments to give more basis for the suggested mechanism. Please include description of this experiment in the Materials and Methods section. As of now, the reactants are mentioned, but the experiment is not described.

Response 6: Thank you very much for pointing this out. In the Materials and Methods section, we have added the materials and methods for the free radical capture experiments (lines 81-83, 146-148, 152-157).

New comment 1: In lines 194-203 the absorption isotherms are discussed. The volume of absorbed nitrogen is merely 20-30 ml (for SiO2 aerogel this value can reach 2 l). This leads no high signal/noise ratio in the pore size distribution curves. It might be better to not provide the image in the text and just mention the cumulative pore volume. If the authors choose to keep this image, please make the y-axis be of the same scale (not 0-20 and 0-30 cm3/g, but both pictures 0-30 cm3/g) and make the caption in the insterted images with pore size distribution readable.

Response 1: Thank you for noticing this and reminding us to make the correction. We have adjusted the Y-axis of the figure and modified the figure caption accordingly (line 224).

New comment 2: Line 208 states: “the smaller the arc radius in the Nyquist diagram, the lower the resistance”, but the radii of the experimentally obtained curves are not given anywhere and the reader is expected to make qualitative conclusions based on the image alone.

Response 2: We agree with that comment. In the Nyquist plot, we compare the relative size of the arc radius of the sample to qualitatively conclude that nitrogen doping reduces the impedance of the sample (line 230).

New comment 3: Line 293 states: “in situ synthesized carbon aerogel possessed 293 more surface functional groups” but the surface groups were not characterized in the work. Use of IR spectroscopy was suggested in my previous review report, but this method was not included. This is up to the Authors, but the conclusion about surface groups seems unbased this way.

Response 3: We recognized that it was not rigorous to directly state the conclusion of " in situ synthesized carbon aerogel possessed more surface functional groups " without providing infrared spectroscopy data as support. In our original research, we did observe that the carbon aerogel maintained its porous structure and speculated that there might be more functional groups on the surface. To be more precise, we have revised the original sentence to " and after carbonization, the in situ synthesized carbon aerogel maintained a porous structure and may possess more surface functional groups." (lines 315-316). This revision retains the conclusion we derived from our research while avoiding making absolute statements. We understand that IR is an effective mean to verify the number of functional groups and will consider using this technique in our future work.

Thank you again for your valuable comments on our manuscript.

Reviewer 2 Report

Comments and Suggestions for Authors

This paper can be accepted after careful checking. 

Author Response

Thank you again for your valuable comments on our manuscript. We have carefully reviewed and revised the manuscript, thank you for your support and understanding!